# A radial map of multi-whisker correlation selectivity in the rat barrel cortex

Luc Estebanez[1,2], Julien Bertherat[1,2], Daniel E. Shulz[2], Laurent Bourdieu[1] & Jean- François Léger[1]

In the barrel cortex, several features of single-whisker stimuli are organized in functional maps. The barrel cortex also encodes spatio-temporal correlation patterns of multi-whisker inputs, but so far the cortical mapping of neurons tuned to such input statistics is unknown. Here we report that layer 2/3 of the rat barrel cortex contains an additional functional map based on neuronal tuning to correlated versus uncorrelated multi-whisker stimuli: neuron responses to uncorrelated multi-whisker stimulation are strongest above barrel centres, whereas neuron responses to correlated and anti-correlated multi-whisker stimulation peak above the barrel–septal borders, forming rings of multi-whisker synchrony-preferring cells.

[1] École Normale Supérieure, PSL Research University, CNRS, Inserm, Institut de Biologie de l'École Normale Supérieure (IBENS), Paris F-75005, France. [2] Unité de Neurosciences, Information et Complexité, UNIC-FRE3693, Centre National de la Recherche Scientifique, Gif sur Yvette, F-91198, France. Correspondence and requests for materials should be addressed to J.-F.L. (email: leger@biologie.ens.fr).

In rodents, the exploration of the environment results in a variety of correlation patterns of multi-whisker deflections, from multi-whisker asynchronous stick and slip events during texture probing, to highly synchronous barrages of whisker deflections during first contacts with edges and object surfaces. In visual and auditory cortices, the spatio-temporal correlation patterns of sensory inputs are key parameters to explain neuronal firing[1]. Similarly, in the barrel cortex, subpopulations of neurons are tuned to specific multi-whisker correlation patterns[2,3].

Functional maps are a prominent feature of primary sensory cortices and constitute a striking mesoscopic signature of the neural circuits that extract the mapped parameter. In the visual system, spatially extensive stimulus properties, including the spatial position and orientation of edge contrasts, are mapped along cortical layers[4,5], but so far the selectivity to higher-order visual statistics has not been mapped. Similarly, in the barrel cortex, single-whisker features, including the direction of whisker deflection[6,7], have been mapped in the barrel cortex, but so far no cortical maps of the tuning to multi-whisker stimulus statistics has been described.

Here we used two-photon fluorescence microscopy to explore the spatial organization of neurons selectively responding to multi-whisker correlations in layer 2/3 of the barrel cortex.

As in deeper layers[3], we observe that neurons have a vast range of possible functional responses, some preferring synchronous deflections of all whiskers (correlated stimulations) and other preferring asynchronous deflections (uncorrelated stimulations). However, in contrast with deeper layers, we found a map associated with these preferences: while responses to uncorrelated stimulations peaked above the centre of the barrels, neurons tuned to correlated stimulations dominated above the edge between barrel and septum. This finding constitutes the first report of a mapping of the tuning to different stimulus patterns of spatio-temporal correlation in a mammalian sensory cortex.

## Results

**Two-photon imaging during multi-whisker stimulations**. To explore the influence of multi-whisker correlations on neuronal activity, we used two sets of multi-whisker stimuli generated by combining unitary whisker deflections of 24 macrovibrissae. Unitary deflections were made of short oscillation-like patterns[3] at four phases (Supplementary Fig. 1a) applied along two orientations (Supplementary Fig. 1b). Deflections were applied either independently to all 24 whiskers (uncorrelated stimulus) or using the same stimulus sequence for all whiskers (correlated stimulus) (Methods, Fig. 1a,b and Supplementary Fig. 1c). By this means, only multi-whisker statistics varied with no change in single-whisker stimulus statistics. Evoked neuronal activity was measured on lightly isoflurane/$N_2O$-anaesthetized rats (Supplementary Fig. 2) using two-photon calcium imaging (Fig. 1c,d and Supplementary Fig. 3). Neurons were accurately localized with respect to the barrel fields after each experiment (Fig. 1e,f). The mean amplitude of the evoked fluorescent transient in the neuron somata was used as a proxy for their spiking activity (Supplementary Fig. 4a), and the mean Z-scores as a measure of the significance of neuronal responses to each stimulus.

**Neuronal selectivity to stimulus correlations**. We first examined whether layer 2/3 neurons were selective to the correlation statistics of 24 whisker deflections as observed in deeper layers of the barrel cortex[2,3,8] (examples in Fig. 1g,h). During uncorrelated stimulation, 26% of neurons (a total of 2,237 neurons were recorded on 25 rats; Supplementary Fig. 5) responded significantly to the deflection of the principal whisker (PW, see Methods) and 12% of the neurons to the closest adjacent whisker.

A significantly larger proportion of neurons responded to correlated stimulations of all 24 whiskers (34% of 2,237 neurons, Fisher exact test $P < 6.4E-9$). When comparing the response level with both types of stimuli, a significantly larger proportion (Fisher exact test $P < 1E-10$, $N = 2,237$ neurons) of neurons preferred correlated (28%, $Z_{corr} > Z_{PW}$) over uncorrelated deflections (8% for the PW, $Z_{PW} > Z_{corr}$). The same trend was present between cells significantly selective to correlated (13% of 2,237 neurons) versus uncorrelated ($<1\%$) deflections (see Methods). Overall, neurons were selective to the degree of multi-whisker correlation in layer 2/3. In contrast with deeper layers, selectivity to correlated inputs dominated in supragranular layers[3].

**Mapping of neurons selective to correlated stimuli**. We then explored the spatial organization of neurons tuned to multi-whisker deflection statistics on top of the barrel map. In individual experiments, whereas responses to the PW during uncorrelated stimuli peaked above the centre of the corresponding barrel, responses to correlated stimulations were maximal in neurons at the periphery of barrels (Fig. 2a, Supplementary Fig. 4b–d and Supplementary Fig. 6). We confirmed this observation (Fig. 2b) by projecting the radial position of all neurons in a normalized radial space (Supplementary Fig. 7a). When data from all experiments were combined (Fig. 2c), we observed the same radial organization of the responses, considering either the mean amplitudes of the fluorescent transient (Fig. 2d), the mean Z-scores (Fig. 2e), the proportion of neurons preferring one of the two stimuli (Fig. 2f) or the proportion of neurons significantly selective to one of the two stimuli (Fig. 2g). These radial profiles produce a concentric map of relative responses to the two types of multi-whisker correlation stimuli (Fig. 2c).

Is this correlation preference map organized in compartments that result from a direct projection of layer 4 barrel and septum domains? We projected the mean responses onto normalized septal coordinates to account for various septum sizes (Supplementary Fig. 7b). This representation showed a significant decrease in the middle of the septa for the correlated-evoked Z-scores (Fig. 2h) and for the proportion of correlation-tuned neurons (Fig. 2i). Correlation-preferring neurons were thus distributed on a ring above each barrel border. This structure thus differs from the map of barrels and septa in layer 4.

**Mapping of neuronal nonlinearity**. Receptive fields are often quantified in the barrel cortex by counting the number of whiskers that evoke a response in a particular neuron, using stimulation of one whisker at a time. Neurons above a barrel predominantly respond to the corresponding whisker, whereas neurons above septa often respond to several adjacent whiskers. Receptive fields are therefore expected to increase along the barrel-to-septum axis[9]. Does the large amount of correlation-preferring neurons that we observed merely reflect an increase of the receptive fields size above the barrel to septa border, or does it correspond to an increase of supra-linear integration of multi-whisker inputs at this position? We computed the nonlinearity (NL) index of each neuron by comparing the sum of its responses to individual whiskers during uncorrelated stimulation, with its response to correlated stimulation (see Methods and three example neurons in Fig. 3a). A majority of layer 2/3 neurons behaved as nonlinear integrators, with a tendency to supra-linearity (Fig. 3b, NL index $>1$ for 76% of correlation-preferring neurons). Supra-linearity was not distributed uniformly across layer 2/3. The radial distribution of NL index for supra-linear neurons exhibited an increase from

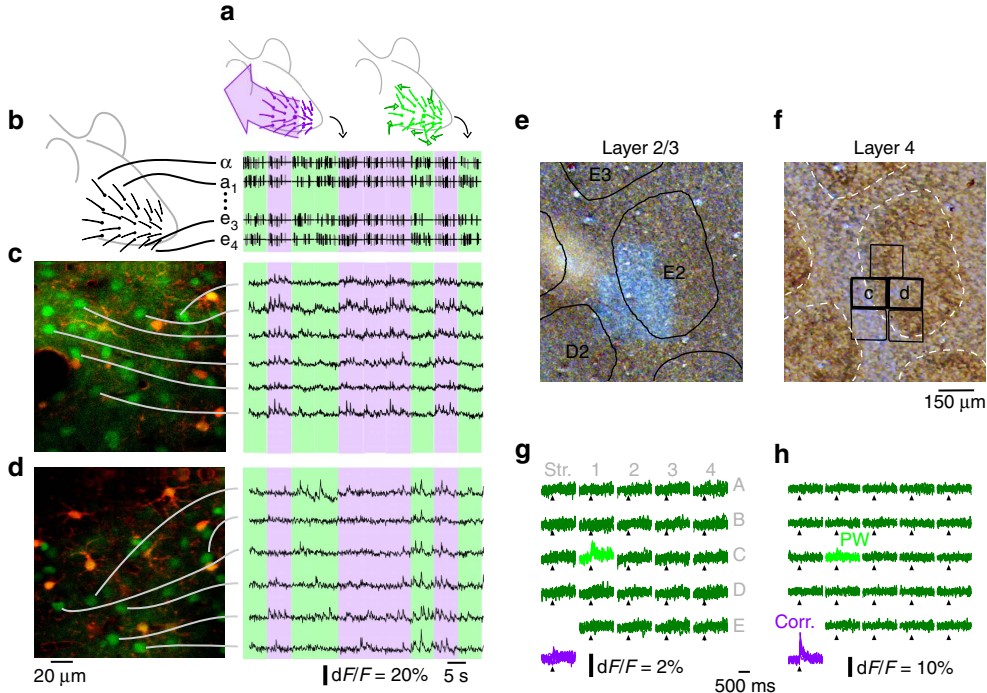

**Figure 1 | Two-photon calcium imaging during multi-whisker stimulations.** (**a**) Cartoon representation of correlated (purple) and uncorrelated (green) multi-whisker stimulations. (**b**) Representative example of multi-whisker stimuli. Purple/green backgrounds: correlated/uncorrelated stimuli. (**c**) Left: example two-photon microscope field of view located above the barrel border. Green: neurons. Red: astrocytes. Right: fluorescence trace of selected neurons. (**d**) Same as **c** for a field of view located inside the area above barrel. (**e**) Histological identification in cytochrome oxidase-stained tangential slices of the barrel cortex of five fields of view acquired next to each others in layer 2/3. Overlay: position and name of layer 4 barrels. (**f**) Corresponding layer 4 slice (barrels inside dashed lines highlight). Overlay: position of the fields of view, including fields displayed in **c** and **d**. (**g**) Representative examples of stimulus-averaged neuronal responses to whisker deflections during uncorrelated stimulations (green, whisker labelled; light green, corresponding anatomical PW) and during correlated deflections (purple). The responses to the different stimulus phase/direction are overlapped. Note the preference for uncorrelated over correlated stimulations (**h**) Same as **g** for a neuron that only responds to correlated stimulations.

barrel centre to border, with a maximum above the barrel–septum border (Fig. 3c), similar to the radial profile of responses to correlated stimuli (Fig. 2d). Finally, the proportion of supra-linear neurons increased from barrel centre to septum, with a peak at the border (Fig. 3d), similar to the radial distribution of the proportion of cells preferring correlated stimulation (Fig. 2f). The origin of the map for correlation-preferring neurons is therefore likely to reside in the uneven distribution of supra-linear integration properties of layer 2/3 neurons.

**Responses to anti-correlated stimulations**. In layers 4–6, some uncorrelation-preferring neurons are facilitated[3] by anti-correlated stimulation of the principal versus all other whiskers (Fig. 4a and Supplementary Fig. 1c). Does such stimulation also trigger strong responses in layer 2/3 and does it show a specific mapping? We found here that unlike in deeper layers, neurons in superficial layers that are selective to uncorrelated stimulation are not facilitated by anti-correlated stimulation (for neurons preferring uncorrelated over correlated stimulation, the median $Z_{anti}/Z_{PW}$ is 0.80, and only 1.3% of these cells have a $Z_{anti} > 2 \times Z_{PW}$). Instead, the maximal responses over all phases and directions to anti-correlated and correlated stimulations were similar (Fig. 4b) and showed the same spatial organization (Fig. 4c). These experiments suggest that the radial map described in Figs 2 and 4 is more than a ring of neurons selective to correlated stimulations, and instead is a ring of neurons selective to the synchronous multi-whisker stimulations, either correlated or anti-correlated.

One should notice that despite the similarity of this ring-shaped map for correlated and anti-correlated responses, these two types of stimuli were not identical in the way they drive neuronal activity. This becomes apparent when comparing the responses in each of the directions and phases explored in our stimulations instead of comparing only the maximum response over all direction and phase conditions. Figure 4d shows a representative neuron, which peak response to correlated stimulations occurs for a pair of phase/direction values that differs from the pair where the anti-correlated response peaked. Among all neurons included in this study, a majority (60%) of responsive neurons exhibit at least one significant difference between their response to correlated and anti-correlated stimuli when phase and direction conditions were compared one by one (bootstrap $P < 0.05$ plus Bonferroni correction, Fig. 4e, 2,237 neurons). The ring of synchronous multi-whisker stimulation-preferring neurons therefore contains a rich diversity of selectivities to various combinations of directions and phases, potentiating its possible importance for complex multi-whisker patterns encoding or treatment.

**A model for the emergence of the maps**. Our study provides evidences for the existence of two maps, for correlated and anti-correlated stimulation-preferring neurons, both with a ring shape above the barrel–septum border. We showed that they spatially correspond to locations where supra-linearity properties dominate, allowing the boost of cell responses to synchronous multi-whisker activation. Why would the supra-linearity be

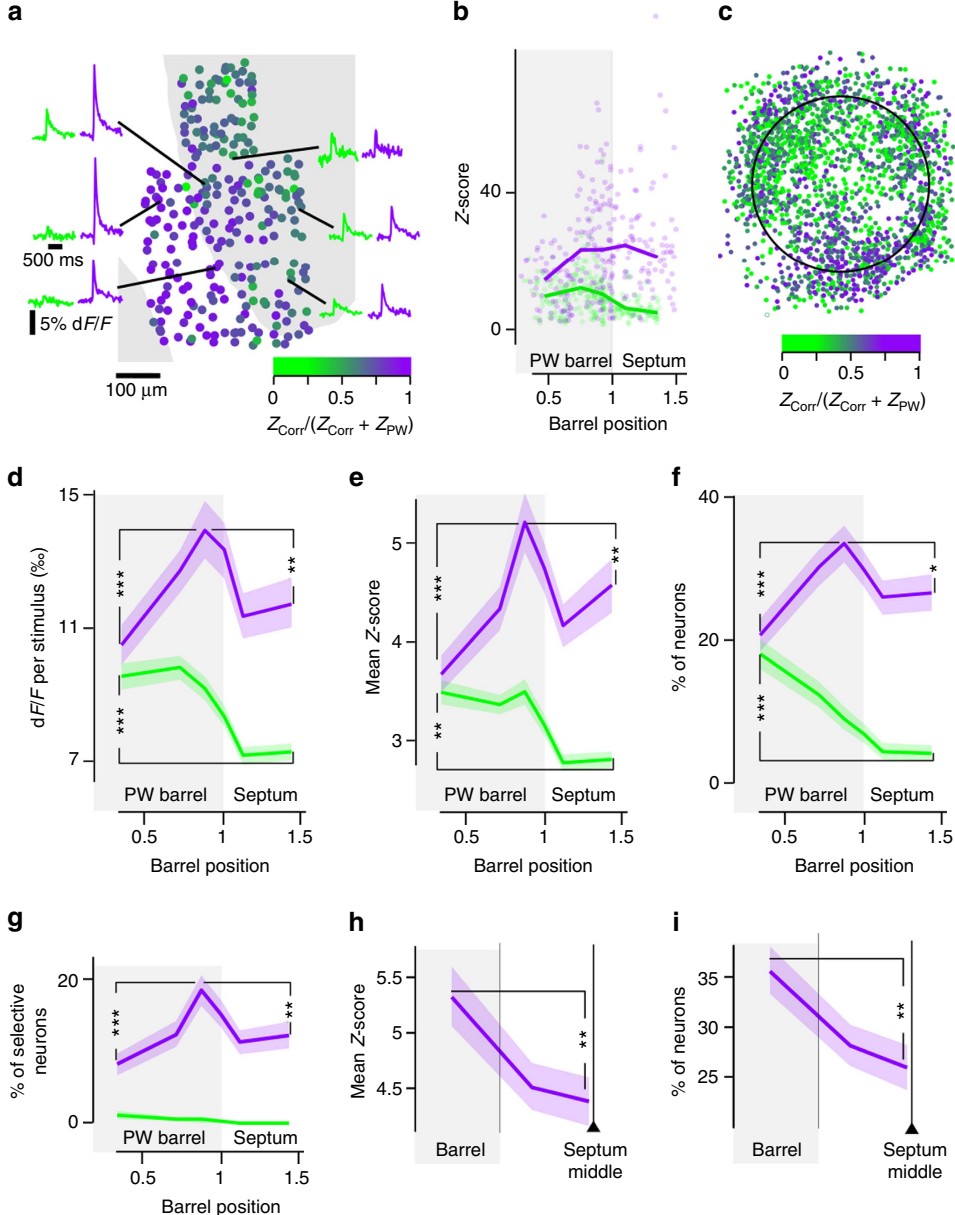

**Figure 2 | A radial map of responses evoked by correlated (purple) versus uncorrelated stimulations (measured on the PW, green).**
(**a**) $Z$-score-derived correlated/uncorrelated stimulation preference index in an individual experiment. Marginal curves: example stimulus-triggered average calcium transients. Grey background: layer 4 barrel. White: septum. (**b**) Projection of $Z$-scores from **a** in normalized barrel radius coordinates. Lines: local $Z$-score median. (**c**) Projection in a normalized barrel of all recorded neurons. Colour code: $Z$-score-derived correlated/uncorrelated stimulation preference index. (**d**) Radial profile (364 neurons per bin) of mean correlated/uncorrelated stimulation-evoked d$F$/$F$, for all recorded neurons. Light background: bootstrap-derived 70% confidence interval. ***Mann–Whitney $P = 5.3E-4$ for uncorrelated and $2.1E-6$ for correlated stimulation-evoked responses. **$P = 2.0E-3$. (**e**) Same as **d** for $Z$-scores. **Mann–Whitney $P = 2.1E-3$ for uncorrelated and $3.6E-3$ for correlated stimulation-evoked responses. ***Mann–Whitney $P = 5.0E-7$. (**f**) Normalized radial profile (364 neurons per bin) of the proportion of neurons preferring correlated (purple) or uncorrelated (green) stimulations, among all responsive neurons. ***Fisher exact $P = 3.5E-9$ for correlated and $1.9E-4$ for uncorrelated stimulation-preferring neurons. **$P = 3.3E-2$. (**g**) Same as **f** for significant tuning to correlated versus uncorrelated stimulations. ***Fisher exact $P = 6.1E-5$. **$P = 8.2E-3$ (364 neurons per bin). (**h**) Correlated stimulation-evoked mean $Z$-scores for all neurons, in normalized septal coordinate (444 neurons per bin). From left to right, equal neuron count bins contain neurons from area above barrel edge, septum edge and septum centre. **Mann–Whitney $P = 2.8E-3$. (**i**) Same as **h** for the proportion of responsive correlation-preferring neurons over all cells. **Fisher exact $P = 2.2E-3$ (444 neurons per bin).

located above the barrel–septum border, building a correlation map that largely escapes the area above barrel? It may arise from integration at multiple stages of the thalamo-cortical loop[10] as well as from local cortical processing. On the basis of our observations, we propose a cortical model of supra-linear integration of multi-whisker inputs. Thalamic inputs to layer

2/3 neurons have three main origins: (i) mono-whisker inputs come from VPm-core in layer 4 barrels, and are then vertically relayed to layers 2/3. (ii) Multi-whisker inputs are conveyed by VPm-head[11] to layer 4 septa, and then again vertically relayed to layers 2/3 (ref. 12). (iii) Multi-whisker inputs can also originate in POm and reach layer 1, where it connects to layer 2/3 dendritic

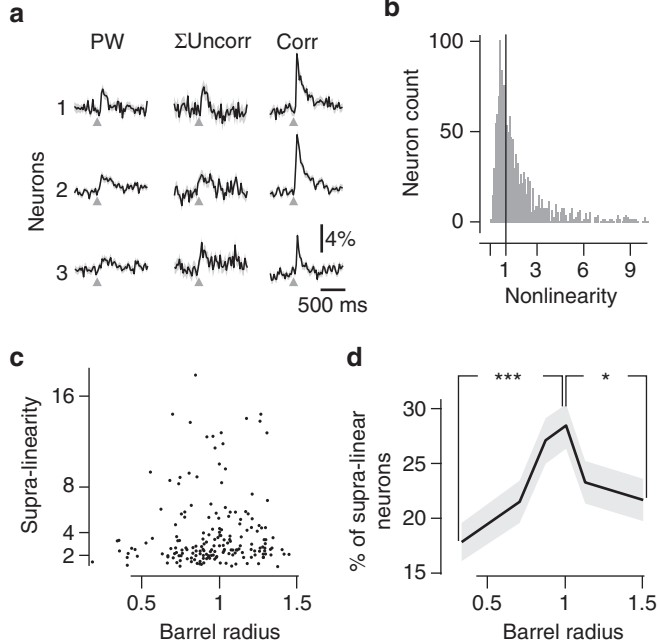

**Figure 3 | Nonlinearity of neuronal responses to correlated multi-whisker stimulations.** (**a**) Three examples of neurons' functional responses to PW stimulation (left), sum of the response to PW and two closest adjacent whiskers (middle), and to correlated stimulation (right). (**b**) Distribution of the NL index. Sixty-eight per cent of neurons are supra-linear. (**c**) Radial distribution of NL for significantly supra-linear neurons (that is, bootstrap-derived response $P < 0.05$). (**d**) The radial distribution (517 neurons per bin) of the proportion of supra-linear neurons (NL > 1), among all responsive neurons, shows a significant peak. *Mann–Whitney $P = 1.4E-2$. ***$P = 9.0E-5$.

arbors. However, recent studies have shown that in anaesthetized animals sensory transmission through this nucleus is often strongly diminished or impeded by feed-forward inhibition from the zona incerta[13,14]. Because of their uneven distribution above barrel and septa, the pathways (i) and (ii) could contribute to the emergence of the map. The spatial profile of the response to deflections of the PW is centred on the corresponding barrel, due to VPm-core inputs. VPm-head inputs centre the response to surrounding whiskers above the septa, as observed in Fig. 5a, where the radial distribution of the responses to PW and to the sum of two closest adjacent whiskers is plotted. Depending on their location, neurons in layer 2/3 could thus have three patterns of input (Fig. 5b): mainly mono-whisker (PW) inputs above barrels; mainly multi-whisker inputs above septa; and a combination of both above the border between barrel and septa. In the case of layer 2/3 neurons above the barrel–septa border, supra-linear summation of PW and surround inputs during correlated or anti-correlated stimulations could then occur, through nonlinear dendritic integration[15,16]. Figure 5b illustrates the expected profile of the supra-linearity. This model predicts a map for correlation-preferring neurons with a ring shape aligned vertically with each barrel border, differing from the simple two-component (barrel and septa) domains, and the existence of a nonlinear cortical integration in the multi-whisker processing that should appear to be non-uniform above barrel and septa. The same mechanism could account for the map for anti-correlation-preferring neurons.

## Discussion

Mappings of stimulus features are typically observed for sensory parameters of perceptual relevance. Here we have shown that temporal correlations in multi-whisker deflections drive efficiently neurons in layer 2/3. Moreover, we report that neurons

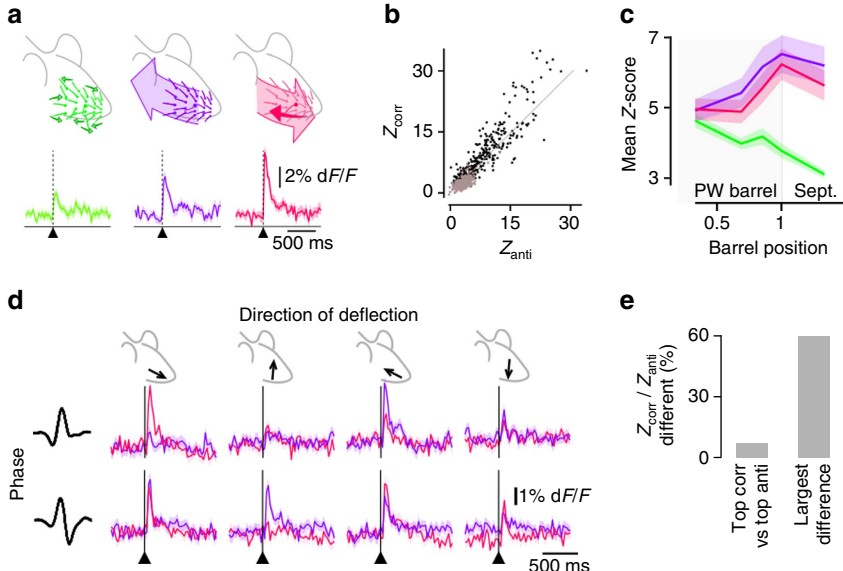

**Figure 4 | Comparison of the functional responses to anti-correlated stimulations with the responses to uncorrelated and correlated stimulations.** (**a**) Top: cartoon representation of uncorrelated (green), correlated (purple) and anti-correlated (pink) multi-whisker stimulations. Bottom: representative example of the average calcium transients triggered by each stimulation for a neuron preferring anti-correlated stimulations. Light background: bootstrap-derived 70% confidence interval of the calcium transient estimate. (**b**) Scatter plot of the Z-score of all neurons in response to correlated PW stimulation ($Z_{corr}$) versus anti-correlated stimulations ($Z_{anti}$). Grey: neurons preferring uncorrelated over correlated stimulations. (**c**) Mean uncorrelated (green), correlated (purple) and anti-correlated (pink) Z-score radial distribution. Through barrel radius, correlated Z-scores are not significantly different from anti-correlated Z-scores. Light background: bootstrap-derived 70% confidence interval of the mean Z-score estimate. (**d**) Responses of a representative neuron to anti-correlated (pink) stimuli across two phases and four directions (rostral, caudal, dorsal and ventral) of stimulation. Same for correlated stimuli, in purple. (**e**) Proportion of responsive cells showing a significant difference between responses to correlated (corr) and anti-correlated stimuli, on their largest response (all phases and directions confounded, left), or at least in one condition of phase and direction out of eight (right). Sept., septum.

 5

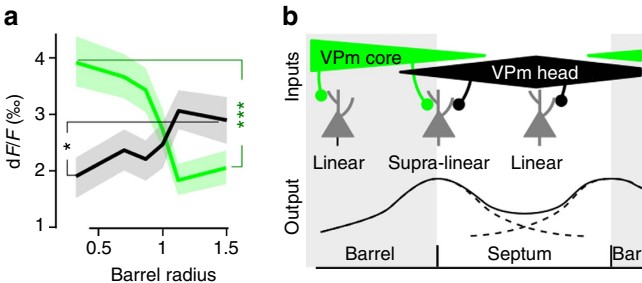

**Figure 5 | A model of multi-whisker supra-linear integration in barrel cortex layer 2/3.** (**a**) Radial distribution of mean dF/F response to the PW (green) and to the sum of the two closest adjacent whiskers (black). ***Mann-Whitney $P = 8.1 \times 10^{-4}$ for PW and *$P = 0.024$ for adjacent whiskers. (**b**) A simple model of the projections to the barrel and septal compartments. A supra-linear integration of VPm-head and VPm-core inputs during correlated stimulation may occur in neurons above the barrel-septa border that receive both inputs, while neurons above barrel centre and septum centre mainly receive a single source of input and are therefore not subjected to the same mechanism. Bar., barrel.

responding to temporal correlations are distributed on rings located at the vertical projections of layer 4 barrel borders. This map superimposes onto several already-known maps for single-whisker stimulation features[6,17,18]. It may arise from integration at multiple stages of the thalamo-cortical loop[10] through lemniscal and paralemniscal pathways, as well as local cortical processing.

Neurons in supra-granular layers of the somatosensory primary cortex are known to exhibit low firing rates, with a large fraction of neurons remaining silent, especially under anaesthetized conditions[19,20]. Here we show that a fraction of these cells considered classically as silent is in fact engaged in the representation of correlation between multiple co-occurring whisker deflections. Among all cells studied under our three multi-whisker stimulus statistics, 41% responded to at least one stimulus and 15% of all neurons were responsive only to correlated or anti-correlated stimuli. These cells would have been counted as non-responsive if tested only with an uncorrelated stimulation of one or a few whiskers, which in our study is the closest to the classical stimuli used to explore barrel cortex receptive fields[7,21,22]. This fraction of multi-whisker-only responsive cells might even be underestimated as our dense stimuli significantly sharpen the receptive field through lateral inhibition[3,8] and could have potentially reduced the number of responding neurons when compared with more classical sparse stimulations.

We show here that many neurons selective to synchronous multi-whisker stimulations exhibit supra-linearity. We cannot exclude that this observed supra-linearity could reflect the low sensitivity of the calcium dye Oregon-Green BAPTA 1-AM (OGB1-AM) to detect single spikes, so that the neuronal supra-linearity would result from supra-linear detection of calcium signal rather than supra-linear emission of spikes. However, this remains unlikely as OGB1-AM, unlike other calcium dyes such as GCamp6, has been reported to show sub-linear rather than supra-linear multi-spike sensitivity[23]. Thus, we suggest that supra-linearity, although observed on the basis of the calcium activity, reflects the network ability to extract multi-whisker synchronous events.

Similar to previous observations obtained in layers 4–6, we found that neurons in layer 2/3 of the barrel cortex have strong responses to multi-whisker stimuli, and many are tuned to specific multi-whisker stimulus statistics. However, differences were also clear between layers. In particular, although neurons responsive mostly to correlated stimulations were recorded across all cortical layers, layer 2/3 neurons were less correlation-selective than in

deeper layers, where we found 'global' neurons that specifically responded to correlated stimulations. Similarly, none of the layer 2/3 neurons that preferred uncorrelated stimulations showed enhanced responses to anti-correlated stimulations, in contrast to deeper layers, where they were referred to as 'antagonist' neurons. These neurons showed overall facilitation by anti-correlated stimulations and suppression by correlated stimulations, while in layer 2/3, neurons preferring anti-correlated stimulations often showed a preference for correlated stimulations at other phases/directions, and did not respond at all to uncorrelated stimulations. These discrepancies support the view that different layers of the barrel cortex perform separate stages of sensory processing[24], and may be responsible for the elaboration of correlated and anti-correlated-tuned multi-whisker receptive fields.

Functional maps are a key organizing principle of sensory and motor cortices. They are observed in low-level cortical areas for simple stimulus features. In higher-order cortical areas, higher-order features are encoded but generally without a clear spatial mapping. In contrast, in our study, we found that spatiotemporal correlations of the stimuli defined at the scale of the whole whisker pad were also spatially mapped onto a primary sensory area, the barrel cortex, on top of its classical whisker-to-barrel somatotopy. It is to our knowledge the first case of mapping of correlation statistics within a primary sensory cortex. Actually our multi-whisker stimuli were far from spanning exhaustively all possible multi-whisker stimulation patterns, and many other non-local multi-whisker stimulus properties could potentially also be mapped into the barrel cortex, such as for instance direction tuning of global apparent motion[25].

## Methods

**Sample size.** Because of our lack of *a priori* knowledge regarding the potential spatial organization of the barrel, we were unable to estimate a sample size accurately. Instead, we accumulated a large pool of data (25 animals total, the activity of 2,237 neurons was acquired, details in Supplementary Fig. 5) and used only non-parametric and exact tests (Fisher exact test, Mann–Whitney test, and bootstrap-based test).

**Animal preparation.** Procedures were in accordance with European and French law and CNRS guidelines. Adult (3–5 months, 300 g) male Sprague Dawley rats without prior experimental history were anaesthetized (isoflurane 2–4%, in 20% $O_2$, 80% $N_2O$). A 3 mm wide craniotomy was opened over barrel cortex (R: − 2.5 mm; L: 5.5 mm) and dura was removed. A silver electrode was positioned in a second craniotomy to monitor microelectrocorticogram (ECoG), and a piezoelectric sensor under the animal monitored breathing. Isoflurane concentration was adjusted (1.5–0.5%) to maintain regular breathing and a desynchronized ECoG (stage III, plane 1–2, Supplementary Fig. 2).

**Whisker stimulation.** We custom-built a 24-whisker stimulator to deflect each whisker independently in arbitrary directions[26]. We selected four stimuli that are the cardinal points of the optimal stimulus 'phase' space that we previously observed in barrel cortex layers 4–6 (ref. 3). We verified the proper playback on whiskers with a laser telemeter (Supplementary Fig. 1a). We applied these stimuli on the rostro-caudal and ventro-dorsal axes (Supplementary Fig. 1b on the 24 largest whiskers on the right snout of the rat (straddlers to column 4, rows A–E, Fig. 1b). Stimulus time series followed a Poisson distribution (time constant: 500 ms; refractory period: 40 ms) and a uniform distribution selected the phases/orientations sequence. We applied such stimuli following three different patterns (Supplementary Fig. 1c). In 'uncorrelated' stimulations different stimulus series occurred on each whisker. For 'correlated' stimulations the same stimulus was applied on each whisker. 'Anti-correlated' were identical to correlated stimulations, except for the anatomical PW that received a stimulus with opposite direction (same orientation but opposite phase). Presentation of the three patterns followed a precomputed randomized order, with an equal mix of uncorrelated and correlated stimuli (14 rats), and anti-correlated stimuli as well in 11 rats.

**Two-photon imaging.** After successful bolus loading of the neuron somata with OGB1-AM[27] a 10 μl drop of sulphorhodamine 101 (SR101 100 μM in external saline) was used to stain specifically astrocytes[28]. Finally, the craniotomy was sealed with a glass coverslip. Rats for which this procedure was not entirely successful were excluded from further investigation steps.

Fluorescence was monitored with a custom *in vivo* microscope coupled with a Mai-Tai (Spectra Physics) mode-locked Ti:sapphire laser (920 nm)[7]. Fields of view

of 150 or 300 μm were scanned at 40 Hz using a combination of a resonant and a galvanometric scanner. Fluorescence photons were collected onto a fibre bundle that split the light equally onto four GaAs photomultiplier (H7421-40, Hamamatsu) used in photon-counting mode and providing together a linear regime up to 25 million photons per s. Recordings were made within this regime.

Activity was recorded between 140 and 300 μm depth. A 60 μm 'reference' stack (1 μm step) was acquired +30 to −30 μm around the initial recording plane for OGB1 and SR101 fluorescence (Supplementary Fig. 3a). Full-frame OGB1 images were then continuously acquired at 40 Hz during ∼1 h and 700 presentations of each phase/orientation multi-whisker stimulation. Large drifts in the Z axis, if any, were manually corrected during the acquisition.

**Identification of the anatomical PW.** The anatomical PW of barrels exposed by the craniotomy was determined at the start of the experiment, either by using ECoG recordings[7] or by running a pilot 10 min recording of full-field average uncorrelated-evoked calcium activity. Histology was systematically performed for *post hoc* confirmation of the PW.

**Histology.** After data acquisition, laser scanning of the field was tuned to an 800 nm wavelength and maximal power during 10 min (350 mW after the objective). This produced a white bleaching mark on the barrel cortex at the recording site after cytochrome oxidase staining (Fig. 1e). The rat was then overdosed with pentobarbital and transcardially perfused with phosphate buffer followed by 4% paraformaldehyde in phosphate buffer. The barrel cortex of the extracted brain was flattened and stored overnight in a solution of 4% paraformaldehyde, between two glass slides separated by 1.5 mm-thick spacers. The 100 μm-thick tangential slices were cut and cytochrome oxidase-stained. On each experiment, a large proportion of the borders of the 24 largest barrels were identified on layer 4 slices (Fig. 1f) and aligned with the layer 2/3 recording sites' bleaching marks using the marks left by penetrative blood vessels. This histological reconstruction was blind to *post hoc* functional response analysis.

**Extraction of calcium transients.** On the difference between the OGB1 and SR101 stacks normalized between 0 and 1, neuron bodies were bright and astrocytes were black. White neuron soma centres were manually pointed (Vaa3d, vaa3d.org), and the soma borders were automatically identified as the radius, where mean intensity dropped below half of the intensity at the centre of the neuron soma in the same plane. The top and bottom neuron limits were set as the first and last planes, where the intensity in the middle plane of the neuron was less than half the intensity at the neuron centre plane. The resulting regions of interest (ROI) were solids of revolution with diameters (Supplementary Fig. 3f) compatible with known pyramidal neurons morphology[29,30].

Once neuron bodies were identified we determined (after binning from 40 to 10 Hz to improve the signal/noise (S/N) ratio and tracking) the *XYZ* stack positions that best correlated frame by frame the OGB1 movie with the OGB stack. This ensured proper attribution of calcium activity to neurons (Supplementary Fig. 3b) over long recordings (made possible thanks to low bleaching; Supplementary Fig. 3e), and removed fluorescence fluctuations due to movements such as heartbeat (Supplementary Fig. 3c,d). In a few recordings with lower S/N ratios, a slower tracking rate (1 Hz) was used to further increase S/N ratio and tracking accuracy to the expense of fast movement correction.

The following post-processing steps were then performed: (1) a distribution of baseline correlation $C_{\text{random}}$ (correlation between spatially randomized images and the stack) was built. When the correlation between a frame and reference stack was below $\text{Mean}(C_{\text{random}}) + 6 \times \text{s.d.}(C_{\text{random}})$, the frame was excluded. (2) The OGB1 fluorescence ($F$) extracted from individual neuron was renormalized by the baseline fluorescence: $dF/F = (F - F_0)/F_0$, with $F_0$ a running average of $F$ on 30 s around current frame. (3) Fast vertical movements could disrupt calcium traces by moving in and out of a neuron region of interest. To remove such noise from traces, the data were split in 5 s epochs that were removed from the data if their s.d. exceeded 0.1 $dF/F$. (4) Only neurons recorded during at least 30 presentations of each stimulus phase/orientation were included in the analysis.

**Analysis of functional data.** Evoked $dF/F$ transient amplitudes were measured in a 100 ms post-stimulus window and normalized by subtracting the baseline activity 500 ms pre-stimulus. Z-scores were derived by division with the s.d. of the 500 ms baseline window. The response to the different stimulus correlation patterns (PW uncorrelated, correlated or anti-correlated) was defined as the largest of the responses to each of the eight dimensions of the stimulus (4 phases × 2 orientations) measured in the given statistic.

In four experiments (11 recordings), spike times were extracted from the OGB1 fluorescence traces using a state-of-the-art peeling algorithm[31] (Supplementary Fig. 4a). The maps derived from Z-scores (Supplementary Fig. 4b) were similar to the maps derived from stimulus-triggered spikes firing (Supplementary Fig. 4d–g).

Neurons included in Fig. 2f,g,i responded significantly to at least one stimulus. To identify neurons with significant stimulus-evoked responses, a threshold on the corresponding Z-score was used. For each neuron, a threshold corresponding to a 5% false-positive rate was estimated based on a distribution of sham scores derived from randomized stimulus times. Since the responses to stimuli statistics were

defined at the maximum of the Z-score measured in response to 8 unitary stimuli (4 phases and 2 directions), the sham scores were the maximum of 8 Z randomized scores. Finally, the shared, population-wide threshold ($Z = 3.9$) was obtained by computing the average of the thresholds obtained for each neuron.

To test if functional responses to correlated versus uncorrelated stimulations were significantly different (Fig. 2g), a bootstrap-based within-cell test was used. For the two conditions, Z-scores were computed for each individual sweep $i$ ($Z_{\text{corr}}(i)$ and $Z_{\text{PW}}(i)$). The amplitude of the difference was computed as the difference of the averaged Z-scores for the two conditions, $\text{mean}(Z_{\text{corr}}(i)) - \text{mean}(Z_{\text{PW}}(i))$. A control random distribution was then obtained by repeatedly (500 times) performing the exact same calculus, but this time on Z-scores drawn with replacement from the mix of all sweeps pertaining to each condition. That is, for one bootstrap repetition, a list of $Z_{\text{corr}}^{\text{Bootstrap}}(i)$ was drawn by randomly selecting elements in the list resulting from the merge of $Z_{\text{corr}}(i)$ and $Z_{\text{PW}}(i)$. The same was done for a second list of $Z_{\text{PW}}^{\text{Bootstrap}}(i)$. This ensured by construction that the distribution of the $\text{mean}(Z_{\text{corr}}^{\text{Bootstrap}}(i)) - \text{mean}(Z_{\text{PW}}^{\text{Bootstrap}}(i))$ reflects the distribution of the neuron in absence of selectivity to condition 1 versus 2. Responses to the two conditions were considered as significantly different when the strength of the difference $\text{mean}(Z_{\text{corr}}(i)) - \text{mean}(Z_{\text{PW}}(i))$ exceeded 1.4 s.d. of this control distribution (a 5% false-positive rate).

Population differences in Z-scores and in $dF/F$ transient amplitudes were tested for significance using Mann–Whitney tests, and differences between proportions of neuronal populations using the Fisher exact test. All tests were two-sided.

To graphically depict an estimate of the robustness of the various population analyses (including mean Z-scores, mean transient amplitude and proportion of neurons preferring a stimulus), the measurements were always presented a non-parametric 70% confidence interval (the light background on all population curves). To obtain this interval for a measurement resulting from $N$ points, we performed the following procedure. Draws with replacement of $N$ points from the data were performed 1,000 times. For each draw, the measurement (for instance the average of the $N$ points) was computed. From this bootstrap distribution the 70% confidence interval of the measurement was defined as the interval between the 15th and 85th percentile of the histogram.

**Normalized barrel–septum coordinates.** Identified neurons were projected into a shared PW barrel–septum radial geometry, with the PW barrel being assimilated to a circle of radius 1. We traced a line connecting the PW barrel centroid and the neuron position. On this line, the barrel radius was the length of the segment up to barrel border. Normalized neuron position was the ratio of the distance from barrel centroid to neuron divided by barrel radius (schematic in Supplementary Fig. 7a). An additional projection was used to focus on the radial profile inside the septum. In this case the position of the neuron was computed as the distance from neuron to the closest barrel border, normalized by the sum of the distance from neuron to PW barrel border and neuron to adjacent whisker (AW) barrel border (schematic in Supplementary Fig. 7b).

**Nonlinearity of neuronal response.** The degree of nonlinearity produced by a neuron was quantified by its NL index $= C(m)/(\text{Uw1}(m) + \text{Uw2}(m) + \ldots)$, where $C(m)$ is the response to correlated stimulus for the phase and direction $m$ for which it is maximal and $\text{Uw1}(m)$, $\text{Uw2}(m)\ldots$ the responses to separate deflections of whiskers 1, 2... for the same phase and direction $m$. Note that to minimize the blurring effect of additive noise in the divisor, we only summed the responses of the PW and the two closest adjacent whiskers. This does not restrict the generality of our results, as <4% of the recorded neurons have a significant response in more than three whiskers (Bootstrap-derived $P < 0.01$ with Bonferroni correction for multiple comparisons).

**Data availability.** The authors declare that the data supporting the findings of this study are available from the corresponding author on request.

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

## Acknowledgements

Funding was provided by ANR (NATACS NR-07-NEUR-025-01, SMIC ANR-12-BSV4-0027 and NEUROWHISK ANR-14-CE24-0019) and by the European Union seventh Framework programme under grant agreement no. 269921 (BrainScaleS). This work has also received support under the programme Investissements d'Avenir, with the references ANR-10-LABX-54 MEMO LIFE and ANR-11-IDEX-0001-02 PSL* Research University. L.E. was partially funded by the Humboldt foundation. We are grateful to Valerie Ego-Stengel for her careful reading of the manuscript.

## Author contributions

L.E., D.E.S., L.B. and J.-F.L. designed the experiments; L.E., J.B. and J.-F.L. performed the experiments; L.E. analysed the data; L.E., D.E.S., L.B. and J.-F.L. wrote the paper.

## Additional information

**Competing financial interests**: The authors declare no competing financial interests.

**How to cite this article**: Estebanez, L. *et al.* A radial map of multi-whisker correlation selectivity in the rat barrel cortex. *Nat. Commun.* **7,** 13528 doi: 10.1038/ncomms13528 (2016).

**Publisher's note**: 

