## [Peer Review File · Nature Communications]

Editorial Note: this manuscript has been previously reviewed at another journal that is not operating a transparent peer review scheme. This document only contains reviewer comments and rebuttal letters for versions considered at *Nature Communications*.

REVIEWERS' COMMENTS:

Reviewer #1 (Remarks to the Author):

I have gone over the revised paper from Dr Leger and colleagues again. They have already addressed my major concerns in the last rounds.

Reviewer #2 (Remarks to the Author):

This study nicely demonstrates a spatial organization of tuning for correlated vs. uncorrelated whisker stimuli in L2/3 of rat somatosensory cortex. This is a significant finding because it has been unclear how complex stimulus parameters are mapped in S1. They find that uncorrelated (i.e. single-whisker) stimuli are represented most strongly within barrel columns, while synchronous multiwhisker stimuli (both correlated and anticorrelated) are represented preferentially in a ring at the barrel-septa boundary. The methods and results are well presented (with one exception - point 1), and the conclusion is appropriate. The model (Fig 5) is not tested in the paper, but is a useful proposal for the field, and I think is a positive addition to the paper.

I was Reviewer 2 on the prior submission and they have addressed all my concerns, including clear analysis showing that L2/3 organization does not simply result from known receptive field properties of barrels vs. septa. I only have two comments on presentation:

1) Is it necessary to use red-green coding to show tuning properties (Fig 2). 8% of your male readers, including me, are red-green color blind, and I cannot evaluate the spatial patterns of tuning shown in Fig 2a-c. Please consider blue-yellow-white or some other color coding scheme.

2) The finding that most L2/3 neurons show supralinear responses to multiwhisker stimuli could represent true neural supralinearity, but it also could reflect low sensitivity of OGB imaging in detecting single spikes. i.e. supralinear detection of Ca signals, rather than supralinear number of spikes. This should be acknowledged in discussion.

Reviewer #3 (Remarks to the Author):

I remain enthusiastic about the paper by Estebanez et al - it is an important contribution to the field of sensory neuronal processing. In particular, it provides an important step forward

in understanding cortical processing of whisker-acquired information. The findings (see my summary in my previous review) are novel, interesting and important.

The key novel findings are

1. L2/3 neurons show different selectivity to inter-whisker correlations
2. They are organized in barrel-related topography – “local” neurons (preferring uncorrelated whisker patterns) at the center and “global” (preferring correlated patterns) at the periphery of each barrel

As far as I know these findings are novel for any sensory modality.

I was happy with the paper already in the previous round, and I think that the current version is ready for publication.

My only suggestion to the authors is to avoid using the term “column” here – “barrel field” and “barrel-septum border” (or alike) would be more appropriate in my eyes, as there is no evidence for a columnar structure beyond layer 4. Anyway, I leave it to the authors to decide, this is a minor issue.

A radial map of multi-whisker correlation selectivity in the rat barrel cortex.

Luc Estebanez, Julien Bertherat, Daniel E. Shulz, Laurent Bourdieu, Jean-François Léger

REBUTTAL

Comments from Reviewers are in italic font, our replies in red

We thank the reviewers for enthusiastic comments. We have addressed the three remaining requests.

Reviewer #1 (Remarks to the Author):

I have gone over the revised paper from Dr Leger and colleagues again. They have already addressed my major concerns in the last rounds.

We thank the Reviewer for many insightful comments through several readings of our manuscript.

Reviewer #2 (Remarks to the Author):

This study nicely demonstrates a spatial organization of tuning for correlated vs. uncorrelated whisker stimuli in L2/3 of rat somatosensory cortex. This is a significant finding because it has been unclear how complex stimulus parameters are mapped in S1. They find that uncorrelated (i.e. single-whisker) stimuli are represented most strongly within barrel columns, while synchronous multiwhisker stimuli (both correlated and anticorrelated) are represented preferentially in a ring at the barrel-septa boundary. The methods and results are well presented (with one exception - point 1), and the conclusion is appropriate. The model (Fig 5) is not tested in the paper, but is a useful proposal for the field, and I think is a positive addition to the paper.

We thank the reviewer for encouraging comments on the article significance and the many questions raised through several readings of the manuscript.

I was Reviewer 2 on the prior submission and they have addressed all my concerns, including clear analysis showing that L2/3 organization does not simply result from known receptive field properties of barrels vs. septa. I only have two comments on presentation:

1) Is it necessary to use red-green coding to show tuning properties (Fig 2). 8% of your male

readers, including me, are red-green color blind, and I cannot evaluate the spatial patterns of tuning shown in Fig 2a-c. Please consider blue-yellow-white or some other color coding scheme.

We apologize for failing to provide color blind friendly figures in the first place. We have addressed this point by updating all figures to a purple/green color scheme that has been confirmed to hold contrast in front of both protanopia and deuteranopia. This was confirmed by using the Adobe Illustrator color blindness proofing system.

2) *The finding that most L2/3 neurons show supralinear responses to multiwhisker stimuli could represent true neural supralinearity, but it also could reflect low sensitivity of OGB imaging in detecting single spikes. i.e. supralinear detection of Ca signals, rather than supralinear number of spikes. This should be acknowledged in discussion.*

We have updated the discussion (p. 7) to acknowledge the possibility that the observed supra-linearity could result from the detection of calcium signal rather than from the spiking activity of neurons. We note, however, that the calcium sensitive dye used in our study, OGB1, is rather suspected to show a sub-linear increase of fluorescence with the number of spikes, as described for instance in Kerr et al., PNAS (2005). 102 (39):14063-14068.

Reviewer #3 (Remarks to the Author):

I remain enthusiastic about the paper by Estebanez et al - it is an important contribution to the field of sensory neuronal processing. In particular, it provides an important step forward in understanding cortical processing of whisker-acquired information. The findings (see my summary in my previous review) are novel, interesting and important.

The key novel findings are□1. *L23 neurons show different selectivity to inter-whisker correlations*□2. *They are organized in barrel-related topography – “local” neurons (preferring uncorrelated whisker patterns) at the center and “global” (preferring correlated patterns) at the periphery of each barrel*□*As far as I know these findings are novel for any sensory modality.*

I was happy with the paper already in the previous round, and I think that the current version is ready for publication.

My only suggestion to the authors is to avoid using the term “column” here – “barrel field” and “barrel- septum border” (or alike) would be more appropriate in my eyes, as there is no evidence for a columnar structure beyond layer 4. Anyway, I leave it to the authors to decide, this is a minor issue.

We thank the Reviewer for enthusiastic comments. We have removed the term « column » from the text as our point was not to describe the existence of any columnar structure in the barrel cortex, but simply to designate the areas in layers 2/3 located to the vertical of barrels or septa. The term « above » has often been added to clarify this point.